# Macrophages Upregulate Estrogen Receptor Expression in the Model of Obesity-Associated Breast Carcinoma

**DOI:** 10.3390/cells11182844

**Published:** 2022-09-12

**Authors:** Daniela Nahmias Blank, Esther Hermano, Amir Sonnenblick, Ofra Maimon, Ariel M. Rubinstein, Emmy Drai, Bella Maly, Israel Vlodavsky, Aron Popovtzer, Tamar Peretz, Amichay Meirovitz, Michael Elkin

**Affiliations:** 1Sharett Institute of Oncology, Hadassah-Hebrew University Medical Center, Jerusalem 91120, Israel; 2Oncology Division, Tel Aviv Sourasky Medical Center, Sackler Faculty of Medicine, Tel Aviv University, Tel Aviv 64239, Israel; 3Department of Pathology, Hadassah-Hebrew University Medical Center, Jerusalem 91120, Israel; 4Cancer and Vascular Biology Research Center, The Rappaport Faculty of Medicine, Technion, Haifa 31096, Israel; 5Hebrew University Medical School, Jerusalem 91120, Israel

**Keywords:** breast cancer, heparanase, obesity, estrogen, estrogen receptor α, macrophages

## Abstract

Breast cancer (BC) and obesity are two heterogeneous conditions with a tremendous impact on health. BC is the most commonly diagnosed neoplasm and the leading cause of cancer-related mortality among women, and the prevalence of obesity in women worldwide reaches pandemic proportions. Obesity is a significant risk factor for both incidence and worse prognosis in estrogen receptor positive (ER+) BC. Yet, the mechanisms underlying the association between excess adiposity and increased risk/therapy resistance/poorer outcome of ER+, but not ER−negative (ER−), BC are not fully understood. Tumor-promoting action of obesity, predominantly in ER + BC patients, is often attributed to the augmented production of estrogen in ‘obese’ adipose tissue. However, in addition to the estrogen production, expression levels of ER represent a key determinant in hormone-driven breast tumorigenesis and therapy response. Here, utilizing in vitro and in vivo models of BC, we show that macrophages, whose adverse activation by obesogenic substances is fueled by heparanase (extracellular matrix-degrading enzyme), are capable of upregulating ER expression in tumor cells, in the setting of obesity-associated BC. These findings underscore a previously unknown mechanism through which interplay between cellular/extracellular elements of obesity-associated BC microenvironment influences estrogen sensitivity—a critical component in hormone-related cancer progression and resistance to therapy.

## 1. Introduction

Breast carcinoma (BC) is the most commonly diagnosed cancer type and the leading cause of cancer-related mortality among women [1]. BC is a highly heterogeneous disease [2,3]. The widely employed classification, referred to as PAM50, classifies BC into five molecular intrinsic subtypes: luminal A, luminal B, HER2-enriched, basal-like, and normal [3,4]. Over 75% of BCs express estrogen receptor α (ER) in >1% of the tumor cells by IHC, and overlap with luminal A and B subtypes [2]. ERα acts as a key driver of BC development, progression and dissemination [5,6,7,8,9,10,11]. Hormonal therapies for ERα-positive (ER+) BC target ERα either directly, by selective ER modulators and down-regulators, or indirectly, by abolishing estrogen production by inhibitors of aromatase, the rate-limiting enzyme in estrogen biosynthesis. Hence, ER expression level (governed by multiple and poorly-understood mechanisms [6,7,8]) is an important determinant in BC tumorigenesis, as well as in the tumor response to hormonal therapy (reviewed in [7,12]).

Obesity is a known risk factor in ER+/luminal BC subtypes [13,14,15,16,17,18]. Furthermore, once diagnosed, obese ER + BC patients (mostly postmenopausal) have worse clinical outcomes than their lean counterparts [13,17,19,20,21,22,23], and evidence is accumulating that the association between obesity and risk/mortality of ER+ BC is stronger [13,24] than earlier estimated [15,25]. A recent large population-based case-cohort study [13] demonstrated that obese women with Luminal A and B (i.e., ER+) BC subtypes had a 1.8− and 2.2− fold increased BC mortality risk, respectively, compared to normal weight women (defined as BMI <25 Kg/m^2^) [13]. Additionally, obesity was strongly associated with aggressive characteristics of luminal tumors (1.7− and 1.8− fold increased risk of stage III/IV disease and grade 3/4 tumors, respectively) [13]. Of note, obesity was not associated with breast cancer-specific mortality among women who had non-luminal (ER−) tumors [13], in agreement with previous reports [14,26,27].

At least 15 different types of cancer have been linked to obesity [15,28]. Thus, the predominant association between obesity and ER+/luminal, rather than ER−/non-luminal, BC subtypes [13,14,26,27] is puzzling. It is often explained by enhanced estrogen signaling, due to increased production of estrogen in chronically-inflamed “obese” adipose tissue compartments, in the context of adverse activation of macrophages (Mϕ) [16,17,18,20,29,30,31,32]. Indeed, Mϕ mobilization is a characteristic feature of obesity-associated BC [33,34]. In fact, Mϕ are regarded as the key immunocyte type within the breast microenvironment, where they adopt maladaptive phenotypes and mediate the BC-promoting action of obesity [16,17,33,34,35]. Among the components of the obese milieu, free saturated fatty acids (SFAs), the levels of which are chronically increased in obesity [36], are well-known for their ability to trigger adverse Mϕ activation via a toll-like receptor (TLR)-dependent mechanism [35,37,38,39]. The occurrence/severity of Mϕ-driven breast adipose tissue inflammation is higher in obese, as compared to normal weight, women [16]. Similar inflammation has been noted in subcutaneous and intra-abdominal visceral fat depots of obese patients/experimental animals [30,40]. It was suggested that inflammatory mediators, secreted by adversely-activated Mϕ, may upregulate expression of aromatase by stromal cells of “obese” adipose tissue, leading to an increase in estrogen levels, and, thus, selectively promoting ER+ BC in obese postmenopausal patients [13,18,41,42].

However, in addition to estrogen, changes in ER expression levels can profoundly affect breast tumorigenesis and BC response to hormonal therapy. Given the recent evidence linking Mϕ to induction of steroid receptor expression in hormone-responsive tumor types other than BC [43,44], we hypothesized that in obesity-associated BC, Mϕ residing in the tumor microenvironment can regulate cellular levels of ER expressed by carcinoma cells, increasing their sensitivity to the hormone and, thus, contributing to enhanced estrogen signaling. Investigating obesity-associated BC in immunocompetent mice, we found that expression of ER by the tumor cells was markedly increased under obese conditions. Moreover, this increase correlated with augmented tumor growth, as well as with the infiltration/activation of Mϕ in tumor and fat tissue of obese mice. In vitro, Mϕ, stimulated by TLR-triggering obese milieu components (i.e., SFA) augmented ERα expression in BC cells of human/murine origin. We also found that augmentation of ERα expression in BC cells by Mϕ was dependent on heparinase, the sole mammalian endoglycosidase enzyme that degrades heparan sulfate proteoglycan chains at the cell surface and is essential for TLR4-mediated Mϕ activation [45,46,47]. Of note, heparanase was recently shown to sustain Mϕ reactivity in several obesity-related [30,48,49] and unrelated [45,50,51], Mϕ-driven inflammatory conditions. Worth mentioning is the fact that, similarly to obesity per se, expression of heparanase correlates with worse outcome in ER+, but not ER−, breast tumors [52]. Collectively, our findings attest to upregulation of ER expression (especially in the face of enhanced production of estrogen in fat depots) as an important, and previously unidentified, mechanism of obesity-accelerated BC progression, likely explaining the dichotomy of the obesity effects on the risk/poorer outcome between luminal (i.e., ER+) and non-luminal (i.e., ER−) BC subtypes.

## 2. Materials and Methods

### 2.1. Clinical Data Analysis

Breast tumor tissue specimens and clinical data from 58 ER+ breast carcinoma female patients were available from the Sharett Oncology Institute, Hadassah Medical Center, Jerusalem. A summary of clinical and pathological characteristics of the study population is shown in Table 1. The use of these data and formalin-fixed, paraffin-embedded breast carcinoma tissues in research was approved by the Human Subjects Research Ethics Committee of the Hadassah Medical Center. Tissue microarray construction was performed as previously described [30,53]. Immunodetection of ERα was performed as in [30,53], using ERα monoclonal antibody NCL-L-ER 6F11 (Novocastra, Newcastle, UK). ERα positivity and staining intensity (scored as weak (=1), moderate (=2), or strong (=3)) was determined in accordance with [54,55] by an expert pathologist (B.M.). Immunodetection of heparanase was performed using polyclonal rabbit anti-heparanase antibody (733) directed against a synthetic peptide (^158^KKFKNSTYRSSSVD^171^) corresponding to the N-terminus of the 50-kDa subunit of the heparanase enzyme [30,53]. The antibody was diluted 1:100 in 10% goat serum in PBS. Control slides were incubated with 10% goat serum alone. Color was developed as described in [30,53], slides were visualized with a Zeiss axioscope microscope and manually read by an expert pathologist (B.M.). To define tumor as heparanase-positive, a cut-off point of 25% immuno-stained tumor cells was chosen, based on an initial overview of the cases, in order to improve signal-to-noise ratios. Cut-off was chosen before any attempt at correlating heparanase expression with the obese status of the patients.

### 2.2. Cell Culture

MCF7 and T47D cells were cultured in DMEM; E0771, 4T1 and THP-1 cells were cultured in RPMI medium, supplemented with 1 mM glutamine, 50 μg/mL streptomycin, 50 U/mL penicillin and 10% fetal calf serum (FCS, Biological Industries) at 37 °C and 8% CO_2_. For estrogen depletion, cells were cultured for 14 days in phenol red-free DMEM supplemented with 5% charcoal-stripped FCS (Biological Industries, Beit-Haemek, Israel), antibiotics, and supplemented with l mM glutamine, prior to estrogen treatment.

### 2.3. Macrophage Isolation and Treatment

Primary mouse macrophages were obtained applying the isolation procedure described in [30,49,51]. For experiments with fatty acids: palmitic (16:0) and stearic (18:0) acid (Sigma-Aldrich, Rehovot, Israel) were diluted in 95% ethanol and conjugated with fatty acid free-BSA at a 2:1 molar ratio before treatment, yielding a stock concentration of 3 mM. For experiments, the fatty acid-BSA solution was further diluted in the culture medium yielding a final concentration of 200 µM. LPS was diluted to a final concentration of 0.1 ng/mL. TAK-242, TLR4 inhibitor (InvivoGen, Toulouse, France) was diluted to a final concentration of 30 µM.

### 2.4. MTS Assay

Cells were seeded in 96-well culture plates in phenol red-free DMEM, supplemented with 5% charcoal-stripped FCS. MTS assay (Promega, Madison, WI, USA) was performed according to the manufacturer’s instructions. Each experiment was performed at least 3 times. Each data point shows the mean of pentaplicate cultures.

### 2.5. Orthotopic Immunocompetent Mouse Model of Obesity-Associated Breast Cancer

Ten week old, female *wt* C57BL/6J mice (Envigo; Jerusalem, Israel) and heparanase-knock out (*Hpse-*KO) mice [56] on C57BL/6J background were fed *ad libitum* high-fat diet (HFD) (Teklad TD.06414, 60% of total calories from fat), or control diet (CD) (Teklad 2018S) for 15 consecutive weeks. In experimental week 12, when HFD-fed animals became obese, E0771 cells were injected orthotopically into the 4th left mammary fat pads of both HFD-fed (obese) and control diet-fed (lean) *wt* and *Hpse-*KO mice (5 × 10^5^ cells per injection). The tumor volume was monitored until experimental week 15. Animals were then sacrificed, tumor tissue samples collected and snap-frozen for RNA/protein extraction.

For tumor irradiation studies tumor-bearing obese and lean mice were anesthetized 7 days before sacrifice and their tumors irradiated conformally to 5Gy, using surface brachytherapy, essentially as described in [57]. Briefly, radiation was delivered to the tumor-bearing mammary gland by using a brachytherapy after-loader (I^192^ Nucletron microSelectron HDR, Veenendaal, The Netherlands); the brachytherapy sleeve was positioned over the tumor (above a 0.5 cm width silicon bolus). The dose distribution calculations were based on CT simulation; the dose of 4.5Gy was calculated to 1.0 cm isodose line, in order to achieve 5Gy to 90% of the tumor volume, while relatively protecting the surrounding normal tissues. The prescribed dose was confirmed by film dosimetry. Variation of dose inside the tumors was estimated to be within ±15% of the prescribed dose. Tumor growth was monitored immediately prior to irradiation and at the indicated times after irradiation. All experiments were performed in accordance with the Hebrew University Institutional Animal Care and Use Committee.

### 2.6. Immunoblotting

Cell and tumor tissue lysates were homogenized in lysis buffer containing 0.6% SDS, 10 mM Tris-HCl, pH 7.5, supplemented with a mixture of protease inhibitors (Roche) and phosphatase inhibitors (Thermo Scientific, Waltham, MA, USA). Equal protein aliquots were subjected to SDS-PAGE (10% acrylamide) under reducing conditions and proteins were transferred to a polyvinylidene difluoride membrane (Millipore, Darmstadt, Germany). Membranes were blocked with 3% BSA for 1 h at room temperature and probed with the appropriate antibody, followed by horseradish peroxidase-conjugated secondary antibody (KPL, Milford, MA, USA) and a chemi-luminescent substrate (Biological Industries, Beit-Haemek, Israel). Band intensity was quantified by densitometry analysis using ImageJ software.

### 2.7. Antibodies

Immunoblot analysis and Immunofluorescence staining were carried out with the following antibodies: anti-F4/80 (AbD Serotec), anti–ERα D6R2W (13258 Cell Signaling), and ab3575 (Abcam, Cambridge, UK), anti-Actin C-2 (sc-8432 Santa Cruz, Santa Cruz, CA, USA), and anti-GAPDH (RPCA Encor, Gainesville, FL, USA).

### 2.8. Immunofluorescence

For immunofluorescence analysis, DyLight 488 goat anti-rat and CyTM3 donkey anti-rabbit (The Jackson Laboratory) antibodies were used as secondary antibodies. Nuclear staining was performed with 1,5-bis{[2-(di-methylamino)ethyl]amino-4,8-dihydroxyanthracene-9,10-dione (DRAQ5) (Cell Signaling Technology). Images were captured using a Zeiss LSM 5 confocal microscope and analyzed with Zen software (Carl Zeiss) and ImageJ software.

### 2.9. Analysis of Gene Expression by Quantitative Real-Time PCR (qRT-PCR)

Total RNA was isolated from cultured cells or snap-frozen tumor tissue samples using TRIzol (Invitrogen, Waltham, MA, USA), according to the manufacturer’s instructions, and quantified by spectrophotometry. Single-stranded cDNA was amplified from 1 μg of total RNA using a qScript cDNA Synthesis Kit (Quanta, Beverly, MA, USA). Real-time quantitative PCR (qRT-PCR) analysis was performed with an automated rotor gene system RG-3000A (Corbett Research, Sydney, Australia). The PCR reaction mix (20 µL) was composed of 10 µL QPCR sybr master mix (Quanta, Beverly, MA, USA), 5 µL of diluted cDNA (each sample in triplicate) and a final concentration of 0.3 µM of each primer. Hypoxanthine guanine phosphoribosyl transferase (HPRT) primers were used as an internal standard. The following primers were utilized:

mouse HPRT sense: 5′-GTC GTG ATT AGC GAT GAT GAA -3′, antisense: 5′-CTC CCA TCT CCT TCA TGA CAT C-3′;

mouse Estrogen Receptor Alpha sense: 5′- GGG CTG ACT TCA CTT ACA TTT C -3′, antisense: 5′- GGA GCA TCT ACA GGA ACA CAG -3′;

human HPRT sense: 5′- GCT ATA AAT TCT TTG CTG ACC TGC T -3′, antisense: 5′- ATT ACT TTT ATG TCC CCT GTT GAC TG -3′;

human Estrogen Receptor Alpha sense: 5′- TGA TGA AAG GTG GGA TAC GA -3′, antisense: 5′-AAG GTT GGC AGC TCT CAT GT -3′;

### 2.10. Statistical Analysis

The results are presented as the mean ±SD unless otherwise stated. *p* values ≤ 0.05 were considered statistically significant. Statistical analysis of in vitro and in vivo data was performed by unpaired Student’s *t*-test. Pearson Chi-Square test was applied to analyze the relationship between heparanase and estrogen receptor alpha expression of breast carcinoma patients, using SPSS software (SPSS Inc., Chicago, IL, USA). All statistical tests were two-sided.

## 3. Results

### 3.1. Increased ERα Expression Levels in Obesity-Associated E0771 Murine Breast Carcinoma

Tumor-promoting action of obesity in ER+ breast tumors is often explained by enhanced estrogen signaling, owing to induction of aromatase expression in “obese” adipose tissue and augmented extragonadal synthesis of estrogen [30,41,42,58]. However, increase in estrogen signaling may also occur due to the upregulation of ER protein levels in BC cells. To test the impact of obesity on ER expression in BC, we applied an immunocompetent orthotopic mouse model of obesity-associated breast cancer, based on murine BC cell line E0771, growing in female syngeneic C57BL/6J mice with high fat diet (HFD)-induced obesity [30,34,59]. E0771 cells express relatively low levels of ERα and are considered as a luminal B subtype of BC [60,61,62]. HFD-fed C57BL6 mice represent a reliable model of diet-induced obesity and related pathological conditions, i.e., inflammation [63], FA accumulation [64], and obesity-accelerated tumor growth [34,59,65] Importantly, female (as opposed to male) C57BL/6J mice are protected against HFD/obesity-induced hyperinsulinemia/hyperglycemia [66], therefore allowing scrutinization of the specific contribution of obesity-related inflammatory events to BC progression.

In agreement with earlier reports [30,34,59,65], following 12 weeks on HFD, the experimental mice became obese, as evidenced by a 38% increase in their body weight, compared to control diet-fed lean mice (*p* < 0.001). E0771 cells were then implanted orthotopically in obese and lean mice, as described in Methods, and tumor growth was monitored for 14 days. Consistent with previous findings [30,34,59,65], growth of E0771 tumors was augmented in obese, as compared to lean, mice (on day 14 tumors from obese mice were 2.4-fold larger than tumors from lean mice, *p* < 0.004). As expected, the extent of Mϕ infiltration, the hallmark of obesity-accelerated breast tumor growth [33,34], was increased 1.9-fold (*p* < 0.03) in tumors grown in obese versus lean mice (Appendix A). Additionally, the obese state attenuated the sensitivity of E0771 tumors to radiotherapy—backbone treatment modality in ER + BC (Appendix A). Given the role of ERα in growth and radio-resistance of BC [5,6,7,8], we examined ERα expression levels in tumors derived from obese versus lean mice. Immunoblot and qPCR analysis of the tumor tissue revealed that expression of the full length 66-kDa ERα protein and mRNA was significantly increased in E0771 tumors growing in obese, as compared to lean, mice (Figure 1A–C). Of note, there was no difference in the levels of 46-kDa ER isoform (known to antagonize the full-length ER−mediated responses in BC [67,68]) between tumors growing in obese versus lean mice.

### 3.2. Macrophages Stimulated by Obese Milieu Components Mediate Upregulation of ERα in BC Cells

Emerging evidence indicates that tumor-infiltrating immunocytes (including Mϕ) are capable of inducing ERα expression in hormone-responsive tumor types [43,69] via cytokine-mediated mechanism. Based on these observations, along with the notion that Mϕ represent the principal pro-cancerous immunocyte population implicated in obesity-accelerated BC tumorigenesis [35,42], and the well-documented increase in Mϕ infiltration in E0771 tumors under obese conditions (References [30,33,34] and Appendix A), we sought to investigate whether Mϕ are capable of modulating ER expression in the setting of obesity-associated BC. To mimic conditions occurring in breast tumor-bearing obese patients/experimental animals (i.e., adverse activation of Mϕ [35,41] due to the presence of increased concentrations of sFA in circulation [36]), murine and human (i.e., E0771, MCF7) ER + BC cell lines were incubated with medium conditioned by primary Mϕ (isolated as described in Methods), either unstimulated or stimulated by SFA (i.e., palmitate, dominant SFA present in fat depots/circulation of female patients [70] and responsible for adverse Mϕ activation under obese conditions [35,37,39,71]. As shown in Figure 2, SFA-stimulated Mϕ-conditioned medium markedly upregulated ERα protein (Figure 2A,B,D,E), as well as mRNA expression (Figure 2C,F) of ESR1 gene, encoding for ERα, in E0771 and MCF7 cells. The effect of SFA-stimulated Mϕ on ER expression was evident even in the 4T1 murine BC cell line (Appendix A), which is often considered ER−negative, but in fact expresses extremely low levels of ERα [72]. Similar effects were observed in MCF7 cells incubated with a medium conditioned by SFA-stimulated human Mϕ differentiated from promonocytic cell line THP-1 [73] (Appendix A).

Since Mϕ activation by SFA under the obese state involves a TLR4-dependent mechanism [35,37,39,71], we next tested the effect of TLR4 inhibition on the ability of SFA-stimulated Mϕ to induce ER expression. As shown in Figure 3A,B, specific TLR4 inhibitor TAK-242 abolished the ability of SFA-stimulated Mϕ to induce ERα expression in E0771 (A) and MCF7 (B) cells, in agreement with TLR4-activating properties of SFA [37,38,39].

Of note, the canonic ligand of TLR4 lipopolysaccharide [LPS] per se is a component of the systemic obese milieu, since it is chronically present (at extremely low levels, typically ranging between 0.01 to 0.2 ng/mL) in the circulation of obese individuals/experimental animals [74,75,76]. This phenomenon, known as ‘metabolic endotoxemia’ [74,75,76], occurs due to impaired intestinal permeability under obese conditions [74,75,76,77,78]. Interestingly, we found that medium conditioned by Mϕ stimulated by LPS at a concentration reflecting metabolic endotoxemia (i.e., 0.1 ng/mL) also upregulated ER in E0771 and MCF7 cells, while pretreatment of Mϕ by TAK-242 prior to LPS stimulation abolished this upregulation (Figure 3C,D), further validating involvement of TLR4-mediated signaling. Since TLR4 could also be expressed by BC cells per se, to exclude the possible effect of carry-over of TLR4 ligand via the Mϕ-conditioned medium, we compared levels of ERα protein in untreated BC cells (E0771, MCF-7) versus those incubated with medium conditioned by LPS (0.1 ng/mL)–stimulated Mϕ, or those incubated with standard medium containing the same concentration of LPS. As shown in Appendix A, ERα protein levels were significantly increased in BC cells treated with medium conditioned by LPS-stimulated Mϕ, as compared to either untreated cells or cells treated directly with LPS, confirming macrophage specific function.

### 3.3. Macrophages Stimulated by SFA Increase the Sensitivity of BC Cells to Estrogen

To verify the functional significance of ERα upregulation, depicted in Figure 2, we next assessed the effect of medium conditioned by SFA-stimulated Mϕ on the proliferative response of MCF7 cells to estrogen. For this purpose, cells were first depleted of estrogen for 2 weeks (as detailed in Methods) and then treated with 17β estradiol in the presence or absence of medium conditioned by either unstimulated or SFA-stimulated Mϕ (Figure 4). As expected, 17β estradiol increased proliferation of MCF7 under all conditions. However, the presence of medium conditioned by SFA-stimulated Mϕ (but not vehicle-stimulated Mϕ) significantly augmented proliferative response to estradiol (Figure 4), consistent with the upregulation of ERα levels (Figure 2). Importantly, stimulated Mϕ also enhanced MCF7 proliferation under conditions of estrogen depletion (i.e., without addition of exogenous estrogen—Figure 4, white bars). In vitro estrogen depletion procedure ensures that only extremely low residual estrogen levels are present in the medium. In addition, this procedure mimics depletion of estrogen that occurs in BC patients treated with aromatase inhibitors. In this context, it is worthy to note that enhanced MCF7 proliferation, triggered by stimulated (as compared to unstimulated) Mϕ, was observed even in the presence of residual levels of estrogen (Figure 4). These data suggest that in the setting of obesity, upregulation of ER levels in BC cells by SFA stimulated Mϕ may enable their proliferative response even to extremely low levels of estrogen, thus providing a molecular/cellular explanation for the observation that obese patients are more likely to experience ER+BC recurrence after treatment with aromatase inhibitors [23] and generally contributing to the worse response to hormonal therapy in BC [13,19,79]. Collectively, these results confirmed the ability of obese milieu component-activated Mϕ to increase the sensitivity of BC cells to estrogen through upregulation of ERα.

### 3.4. Mϕ-Mediated Augmentation of ER Expression in BC Is Dependent on Heparanase, the Endoglycosidase Enzyme Essential for Mϕ Reactivity

Heparanase enzyme is preferentially expressed in obesity-associated breast tumors in clinical/experimental settings [30] and associated with worse prognosis in ER+, but not ER− BC [52]. Heparanase is the only known mammalian endoglycosidase capable of cleaving heparan sulfate (HS) at the cell surface and ECM. Of note, intact extracellular HS was shown to inhibit TLR4 signaling and Mϕ activation, while enzymatic cleavage of HS relieves this inhibition [80]. Furthermore, soluble fragments released through enzymatic cleavage of HS stimulate TLR4 signaling [80,81,82,83,84]. Thus, heparanase enzyme is causally involved in Mϕ activation by TLR4 ligands [45,46,47,48,50], including FA [30,49]. In light of the above data, we next investigated the role of heparanase in Mϕ-mediated induction of ER under obese conditions.

First, utilizing the obesity-associated E0771 BC model in heparanase-null *Hpse*-KO mice, we found that, unlike in wild-type (*wt*) animals, the obese state failed to upregulate ERα expression in BC tumors under conditions of heparanase deficiency (Figure 5A–C). Of note, heparanase deficiency has been previously shown by Hermano et al., to abolish BC-promoting action of obesity in vivo [30]. Data from these experiments were used in the present study for comparative purposes. It should be noted that heparanase deficiency did not affect growth of E0771 tumors in lean mice (as no statistically significant difference was detected between volume of tumors derived from *wt* lean (105 mm^3^) versus *Hpse*-KO lean (154 mm^3^) mice on day 14, *p* > 0.06). In agreement with the in vivo findings (Figure 5), in vitro Mϕ derived from heparanase-null mice (KO-Mϕ), differently from Mϕ derived from *wt* mice (*wt*-Mϕ), failed to upregulate ERα in murine (E0771, Figure 6A) and human (T47D, Figure 6B) BC cell lines following SFA stimulation. On the other hand, in vitro treatment with recombinant heparanase enzyme prior to stimulation by SFA (mimicking overexpression of the enzyme in obesity-associated breast tumors [30]) resulted in a 4-fold increased ability of SFA-stimulated Mϕ to upregulate ERα expression in E0771 and MCF7 cells (Appendix A). Corroborating the above findings in the clinical setting, immunohistochemical analysis of ER + BC tissue specimens derived from 58 patients revealed that heparanase positivity significantly correlated with higher levels of ERα expression (quantified as in [54] in the tumor tissue): a nearly 2-fold higher proportion of high ERα expression was detected in heparanase-positive versus heparanase-negative tumors (73% vs. 40%, chi-square test *p* = 0.0135, Figure 7).

Of note, although heparanase is expressed by both Mϕ [46,47,48,49] and BC cells [52] numerous observations have repeatedly indicated that in the setting of breast tumors, carcinoma cells per se appear to represent the major cellular source of the enzyme (reviewed in [52]). Given the above notion, the secreted nature of heparanase, and the role of extracellular heparan sulfate in modulation of TLR signaling ([80,81,82]), it is plausible that in the BC under obese conditions the excessive enzyme overexpressed by the tumor cells is a main contributor to adverse activation of macrophages. Collectively, these data highlight a novel mechanism through which interplay between elements of the obesity-associated BC microenvironment, both cellular (i.e., Mϕ, upregulating ERα expression; carcinoma cells, supplying heparanase) and extracellular (obese milieu components, HS), influences estrogen sensitivity—a critical component in hormone-related cancer progression and resistance to endocrine therapy.

## 4. Discussion

Breast tumors remain a major cause of morbidity/mortality worldwide, despite remarkable progress in the diagnosis/endocrine therapy of ER + BC [28]. Approximately 30% of patients with hormone-responsive BC develop resistance. Moreover, the alarming global increase in the prevalence of obesity [85,86], a well-characterized risk factor for both incidence and worse prognosis of ER+/luminal BC subtypes, threatens to limit the progress in the management of this disease, highlighting the need for further in-depth characterization of obesity related ER−driven tumorigenesis.

The diverse spectrum of obesity effects in mammary tumorigenesis, as well as association between obesity and ER+ tumors predominantly in postmenopausal patients [13,14,26,27] (in which circulating levels of the hormone drop after cessation of ovarian estrogen production), are far from being fully characterized [41,42]. The ER+BC promoting action of obesity, mainly in postmenopausal women, has been attributed to increase in circulating [87] and adipose tissue levels of estrogen, as a result of upregulation of aromatase enzyme in fat depots of obese patients [13,16,18,30,41,42]. Yet, postmenopausal circulating levels of the hormone in these patients appear to be insufficient to activate signaling through ERα expressed in the uterine tissue [88] (as endometrial thickening does not occur in the majority of the cases [42]). This notion implies that, in addition to increased estrogen levels, the extent of ERα expression may serve as an important determinant of estrogen-driven BC progression under obese conditions.

Here we showed that tumor-associated macrophages (TAMs) can augment estrogen signaling in BC cells through upregulation of ERα expression. We found that the obese state upregulated ERα expression in a murine model of obesity-associated BC (Figure 1), and demonstrated contribution of Mϕ, adversely-activated by obesogenic substances (e.g., SFA) through TLR4-dependent mechanism, to ERα upregulation in mouse/human BC cells (Figure 2 and Figure 3; Appendix A). We also showed that Mϕ-mediated increase of ERα levels in BC cells influenced estrogen sensitivity (Figure 4)—a critical component in hormone-related cancer progression and response to endocrine therapy [7]. Indeed, ERα is the key driver of breast tumorigenesis in ~75% of cases, and, at the same time, higher ERα load is often considered as predicting better prognosis in the general population of systemically-treated ER + BC patients, which is typically explained by the ability to effectively target ERα by existing hormonal therapy. Noteworthy is the fact that a recent study based on a cohort of >30,000 BC patients found no clear evidence for an association between higher ER load and better outcome [89]. In this respect, our present findings suggest that in the sub-population of obese postmenopausal ER+BC patients (especially in the face of increased estrogen production in fat depots [13,16,20,29,32]), upregulation of ERα expression by the tumor may promote BC progression and render it less sensitive to standard therapeutic approaches (i.e., estrogen deprivation (Figure 4), and ionizing radiation (Appendix A)).

Mϕ infiltrating the tumor tissue or adjacent adipose tissue have been shown to contribute to several important features of malignant growth, and correlate with poor prognostic signs (i.e., high tumor grade, high mitotic activity, and metastasis [17,90,91,92]). Yet, the emerging role of Mϕ (and other tumor-residing immunocytes) in the regulation of steroid receptor expression in hormone-responsive tumors (i.e., through secretion of cytokines capable of inducing ERS1 gene expression) only recently came to be appreciated [43,44,69]. When considered in the context of hormone-dependent tumors, our findings suggest a bidirectional action of Mϕ in obesity-associated breast tumorigenesis, through which TAMs augment estrogen signaling in BC via upregulation of ERα (present study), while adipose tissue Mϕ contribute to this augmentation through an increase in estrogen levels (via induction of aromatase expression in stromal cells of ‘obese’ adipose tissue [16,30,41,42]). Interesting to note, Mϕ, per se, reportedly express ER and estrogen can affect some of the macrophage functions; yet, the role of ER signaling in the control of the Mϕ reactivity remains poorly understood, and both findings supporting the activating and suppressing effects of the hormone on Mϕ have been published [93].

Taken together, the above findings suggest beneficial effect(s) of approaches aimed at manipulating Mϕ in BC, especially when taken together with previous reports attesting to Mϕ being an attractive target in mammary tumorigenesis, in general, and obesity-accelerated BC progression, in particular [16,30,33,34,94]. Yet, initial clinical studies (utilizing various agents decreasing Mϕ content) demonstrated mixed results [95,96]. The likely explanation for the somewhat limited benefit of the direct Mϕ targeting approach is provided by the dynamic nature and highly heterogeneous phenotypes of Mϕ [92,97]. This notion is particularly relevant in the context of obesity: a recent report revealed that ‘obese’ visceral adipose tissue contains seven distinct Mϕ populations with potentially different functions/cytokine profiles [98]. Given this complexity, our findings pointing to the ability of heparanase to dictate the ER−inducing phenotype of Mϕ in BC (Figure 5 and Figure 6), and, therefore, control one of the fundamental features of hormone-dependent breast tumorigenesis, namely, the level of ER expression, suggest that modulation of the enzyme activity (rather than targeting Mϕ per se [95]) may offer an appealing alternative approach to uncouple obesity and breast cancer progression in a rapidly growing population of obese patients. Of note, a recent study (based on microarray datasets comprising >7000 BC patients, as well as prospective data from the BIG 2–98 repository [52]) demonstrated that overexpression of heparanase in BC tumors was associated with worse outcome/increased risk of recurrence in luminal (ER+) subtypes, but did not affect outcome/risk in non-luminal (ER−) subtypes [52]. This pattern of association is essentially similar to one reported for obesity, per se, which correlates with worse prognosis in BC patients bearing luminal tumors, while no such association was found in non-luminal tumors [13,14,26,27], and further supports the contribution of the enzyme to BC-promoting effects of obesity through Mϕ-mediated upregulation of ER.

Collectively, the data presented here reveal previously unknown crosstalk occurring between tumor, adipose and immune compartments in hormone-responsive obesity-associated BC. Moreover, as techniques to effectively screen for the Mϕ infiltration/heparanase expression in tumor tissue become available, and heparanase-targeting approaches are pre-clinically and clinically developed [99], our findings provide a basis for further studies aimed at manipulating estrogen signaling and suppressing breast cancer-promoting consequences of excess adiposity in an increasingly obese population.

## Figures and Tables

**Figure 1 cells-11-02844-f001:**
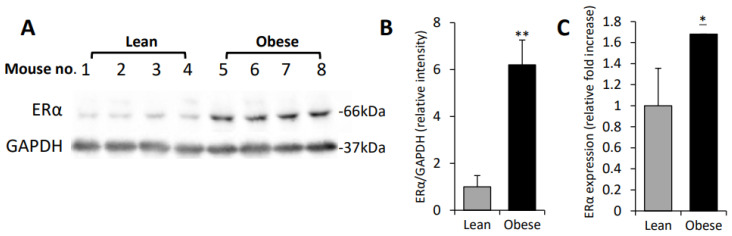
**Upregulation of ERα in obesity-associated murine breast carcinoma model.** Female C57BL6 mice fed for 12 weeks with either HFD (obese) or control diet (lean), were injected with E0771 cells into the 4th mammary fat pad of all mice, as described in Methods. Tumors were harvested 14 days post cell injection and lysates of tumor tissue derived from lean and obese mice were analyzed for ERα expression by immunoblotting and qRT-PCR. (**A**) Protein levels of full length 66 kDa ERα in the orthotopic E0771 tumors derived from lean and obese. (**B**) The band intensity was quantified using ImageJ software; intensity ratio for ERα/GAPDH is shown, *n* ≤ 4 mice per condition. (**C**) qRT-PCR was used to determine the levels of ERα mRNA in tumor tissue samples collected from lean and obese mice. Error bars represent ± SD. * *p* < 0.04; ** *p* < 0.001.

**Figure 2 cells-11-02844-f002:**
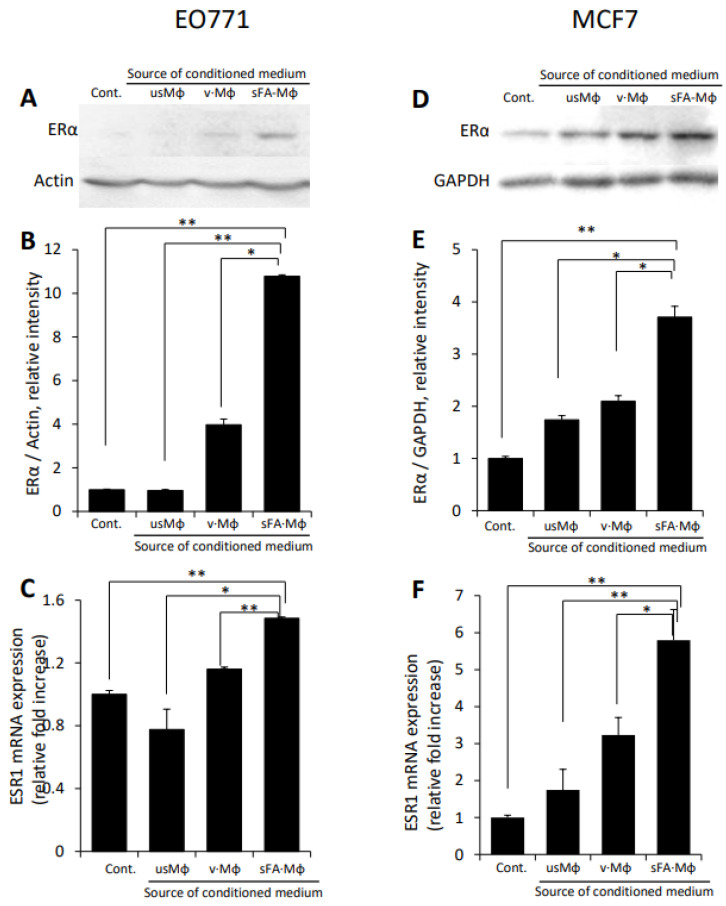
**Upregulation of ER****α expression in murine and human BC cells by SFA-stimulated macrophages.** E0771 (**A**–**C**) and MCF-7 (**D**–**F**) BC cells either remained untreated (Cont.) or were incubated (24 h, 37 °C) with medium conditioned by either unstimulated macrophages (usMϕ), or macrophages stimulated (16 h) by vehicle (BSA) alone (v·Mϕ) or by SFA (i.e., palmitic acid) conjugated with fatty acid free-BSA at a 2:1 molar ratio (SFA·Mϕ). (**A**,**D**) ERα protein levels were determined by immunoblotting. (**B**,**E**) The band intensity was quantified using ImageJ software; intensity ratio for ERα/actin or ERα/GAPDH is shown. (**C**,**F**) The expression of mouse (**C**) and human (**F**) ESR1 gene, encoding for ERα, was determined by qRT-PCR. The data shown are representative of ≥3 independent experiments; error bars represent ± SD. * *p* < 0.02; ** *p* < 0.0015.

**Figure 3 cells-11-02844-f003:**
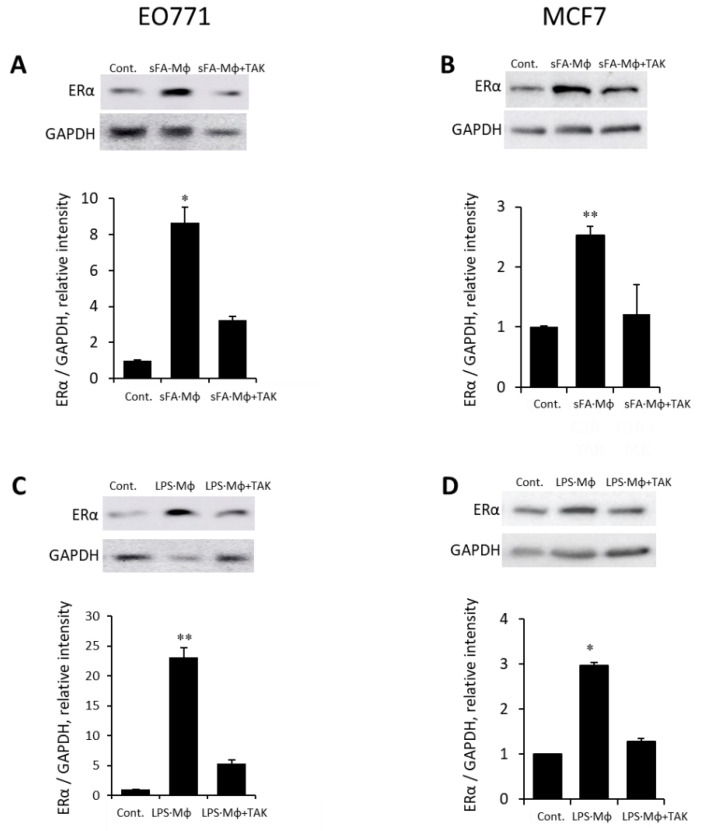
Ability of macrophages to upregulate ER expression in BC cells in response to stimulation by obese milieu components depends on TLR4 signaling. (**A**,**B**). Mouse E0771 (**A**) and human MCF7 (**B**) BC cells either remained untreated (Cont.) or were incubated (24 h, 37 °C) with medium conditioned by macrophages stimulated with 200 µM SFA (i.e., palmitic acid), alone (SFA·Mϕ) or in the presence of 30 µM of TLR4-specific inhibitor TAK-242 (SFA·Mϕ + TAK). ERα protein levels were determined by immunoblotting (top) and quantified using ImageJ software (bottom); the intensity ratio for ERα/GAPDH is shown. (**C**,**D**). Macrophages stimulated by conditions mimicking metabolic endotoxemia upregulated ERα expression in BC cells. E0771 (**C**) and MCF7 (**D**) cells were either remained untreated (Cont.) or were incubated (24 h, 37 °C) with medium conditioned by macrophages stimulated by 0.1 ng/mL LPS alone (LPS·Mϕ) or in the presence of 30 µM of TLR4-specific inhibitor TAK-242 (LPS·Mϕ + TAK). ERα protein levels were determined by immunoblotting (top) and quantified using ImageJ software (bottom). The intensity ratio for ERα/GAPDH is shown. The data are representative of 3 independent experiments; error bars represent ± SD. * *p* < 0.007; ** *p* < 0.0009.

**Figure 4 cells-11-02844-f004:**
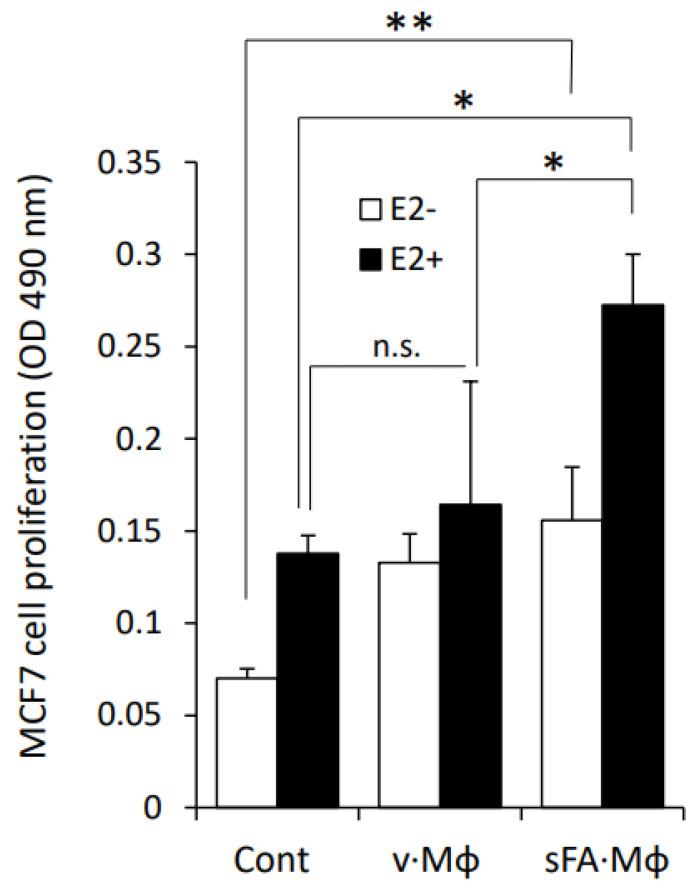
**Effect of SFA-stimulated macrophages on estrogen sensitivity of MCF7 cells.** Prior to estrogen treatment, MCF7 cells were maintained for 2 weeks in phenol red–free medium supplemented with charcoal-stripped FCS (5%), and then incubated (96 h, 37 °C) with the medium conditioned by either unstimulated macrophages (Cont), or macrophages stimulated (16 h) by vehicle (BSA) alone (v·Mϕ) or by 200 µM palmitate (SFA) conjugated with fatty acid free-BSA at a 2:1 molar ratio (SFA·Mϕ), in the absence (white bars) or presence (black bars) of 10^−9^ M estradiol (E2). Cell proliferation was analyzed by MTS assay, * *p* < 0.02, ** *p* < 0.003; n.s: no statistical difference. Error bars represent ± SD.

**Figure 5 cells-11-02844-f005:**
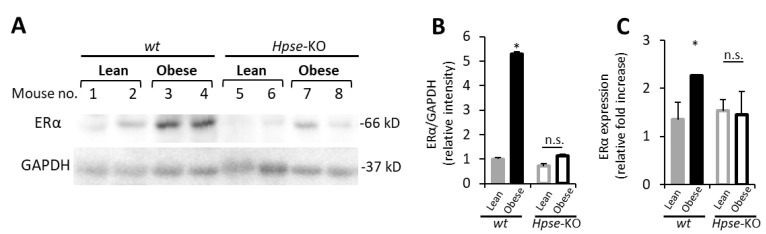
**Heparanase deficiency abolishes effect of obesity on ER****α expression in the murine model of BC.** Wild-type (*wt*) and heparanase-null (*Hpse−*KO) female C57BL6 mice were made obese by administration of HFD for 12 consecutive weeks, as described in Methods. Control (lean) mice were feed with a regular diet. HFD-fed animals of both *wt* and *Hpse*−KO genotypes became obese, as evidenced by their significantly increased body weight compared to control diet-fed lean mice (*wt*: 38% increase *p* < 0.001; *Hpse*−KO: 49% increase, *p* < 0.03). Then, all mice were injected with E0771 cells into the 4th mammary fat pad, as described in Methods. Tumors were harvested 14 days post cell injection and lysates of tumor tissue were analyzed for expression of ERα protein (**A**,**B**) and mRNA (**C**). (**A**,**B**) Levels of full length 66 kDa ERα protein in the orthotopic E0771 tumors derived from lean and obese mice, were analyzed by immunoblotting (**A**). The band intensity was quantified using ImageJ software; intensity ratio for ERα/GAPDH is shown, *n* ≤ 5 mice per condition (**B**). Error bars represent ± SD; * *p* < 0.0002; n.s: no statistical difference. (**C**) qRT-PCR was used to determine the levels of ERα mRNA in tumor tissue samples collected from lean and obese mice.

**Figure 6 cells-11-02844-f006:**
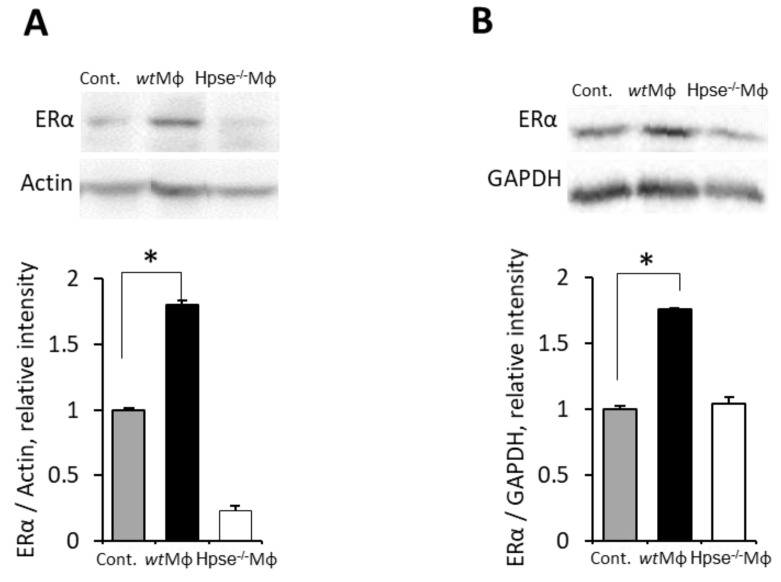
**Heparanase-deficient macrophages stimulated by SFA failed to upregulate ERα levels in BC cells.** E0771 (**A**) and T47D (**B**) cells either remained untreated (Cont.) or were incubated (24 h, 37 °C) with medium conditioned by macrophages derived from either wild-type (wtMϕ) or heparanase-knockout (Hpse-/- Mϕ) mice, and stimulated by saturated fatty acid, either palmitic (**A**) or stearic (**B**), as described in Methods. ERα protein levels were determined by immunoblotting (top) and quantified using ImageJ software (bottom). Error bars represent ± SD. * *p* < 0.002.

**Figure 7 cells-11-02844-f007:**
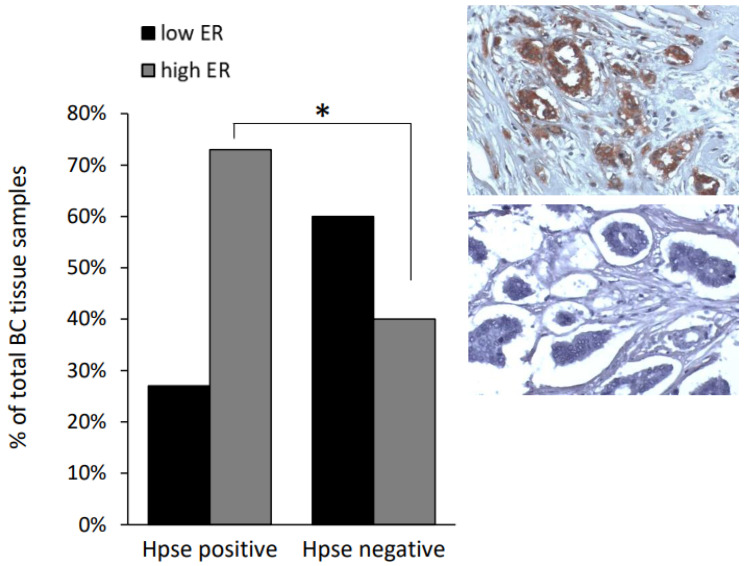
Sections of human ER + BC tissue samples (*n* = 58) were processed for immunohistochemistry with anti-ERα and anti-heparanase (Hpse) antibodies, as described in Methods. **Inset:** representative images of heparanase-positive (top) and heparanase-negative (bottom) breast tumor tissue specimens (invasive ductal carcinoma), original magnification ×200. To define tumor as Hpse-positive, a cutoff point of 25% immuno-stained tumor cells was used, as in Ref. [53]. The intensity of ERα staining was scored as weak (1), moderate (2), or strong (3), as described in Methods; tumors with staining score 1 were categorized as “low ER” (black bars); tumors with staining score 2 and 3 as “high ER” (grey bars). Chi-squared analysis was then used to assess the relationship between heparanase positivity and high versus low ER levels. Significant correlation between expression of heparanase and high ER levels was noted:an almost 2-fold higher proportion of high ER expression was detected in heparanase-positive versus heparanase-negative tumors (73% vs. 40%, chi-square test * *p* = 0.0135).

**Table 1 cells-11-02844-t001:** Clinical and pathological characteristics of the study population (*n* = 123 subjects).

Criteria	Range	No. Cases (%)
**Tumor size (cm)**	0.5–10.5	
<2		31 (25.2)
2–5		50 (40.7)
>5		3 (2.4)
Unknown		39 (31.7)
**Tumor type**		
Ductal		109 (88.6)
Lobular		5 (4.1)
Others		9 (7.3)
**Lymph node status**		
LN pos		38 (30.9)
LN neg		37 (30.1)
Unknown		48 (39)
**Grade**		
1		19 (15.5)
2		46 (37.4)
3		56 (45.5)
Unknown		2 (1.6)
**ER status**		
Negative		40 (32.5)
Positive		78 (63.4)
Unknown		5 (4.1)

## Data Availability

All data generated or analyzed during this study are included in this article.

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
