# Peer review of "Macrophages Upregulate Estrogen Receptor Expression in the Model of Obesity-Associated Breast Carcinoma"

_cells, 2022, doi:10.3390/cells11182844_

Round 1

Reviewer 1 Report

I have been asked to review the original article by Blank et al. with the following title; "Macrophages regulate estrogen receptor expression in the model of obesity-associated breast carcinoma." The study is novel, and the significant contents could benefit all interested in macrophages. In my opinion, the article could be published after minor revisions;

1- I would like to suggest adding the overexpression of ER by the macrophages in your study and changing the title in a way to b more informative and according to your findings. 

2- The keywords could be updated and some keywords such as ERa could be added. 

3- I would like to see some phrases in the introduction explaining the novel targeted therapies for breast cancer that utilize ER. 

4- Some articles could be used to enrich the introduction and discussion, including; PMID: 19372199

5- I think it would be beneficial to talk about the agonists and antagonists of the ERa and their interaction with macrophages.

6- In the methods, since you used the BMDMs from macrophages, why didn't you use monocyte-derived macrophages from humans instead of THP-1?

Author Response

1- I would like to suggest adding the overexpression of ER by the macrophages in your study and changing the title in a way to b more informative and according to your findings. 

RESPONSE: we thank the Reviewer for this suggestion; As recommended, we adjusted the title of the revised manuscript, so that it now reads: "Macrophages upregulate estrogen receptor expression in the model of obesity-associated breast carcinoma."(lines 2, 3)

2- The keywords could be updated and some keywords such as ERa could be added. 

RESPONSE: As requested, we updated the keywords and added ER (line 45).

3- Some articles could be used to enrich the introduction and discussion, including; PMID: 19372199

RESPONSE: as suggested, article PMID: 19372199 is cited throughout the revised manuscript (new reference #18, cited on lines 60, 76, 89 and line 466).

5- I think it would be beneficial to talk about the agonists and antagonists of the ERa and their interaction with macrophages.

RESPONSE: We thank the reviewer for raising this important point. Indeed, macrophages express estrogen receptors and estrogen reportedly affects a variety of

macrophage functions; However, the role of estrogen/estrogen receptor pathways in control of the macrophage reactivity (as well as their survival) is only partially understood. Moreover, both findings supporting activating and suppressing effects of the hormone on macrophages have been published, so the entire topic remains somewhat contradictory (Campbell, L et al., J Invest Dermatol, 2014, 134:2447-2457; Pepe et al., Hum Reprod Update, 2018, 24:652-672; Taneja , V. Front Immunol 2018, 9:1931; Subramanian et al., J Immunol 2007, 179:2330-2338). In light of this complexity, we discuss rather in brief macrophage responsiveness to ER agonists/antagonists (lines 501-504 and new Ref. 93 in the revised manuscript).

6- In the methods, since you used the BMDMs from macrophages, why didn't you use monocyte-derived macrophages from humans instead of THP-1?

RESPONSE: The reason for using THP1-derived macrophages instead of monocyte-derived macrophages is the fact that human monocyte-derived macrophages are short-lived, finite and vary from donor to donor (Baxter EW, et al.,. J Immunol Methods 2020, 478:112721, DOI: 10.1016/j.jim.2019.112721). Thus, THP1-derived macrophages are often considered preferable over primary human monocyte-derived macrophages for the reproducible in vitro experimentation.

Reviewer 2 Report

In their manuscript, Blank et. al. presents a critical role for TAMs in obesity mediated upregulation of ER expression in the hormone responsive breast tumor. The author also identified the role of heparinase in regulating TAMs mediated ER expression in the breast tumor cell lines. The conclusions reached by the authors are supported by the data presented. However, there are a few concerns that needs to be addressed. 

The writing style is not clear. Poorly written, Lengthy sentences and too wordy.

Do the authors notice any significant difference in tumor growth and ER expression between lean WT and lean hpse-/- mice?

What figure or supplementary data does the authors referring to in following section?  “On the other hand, in vitro treatment with recombinant heparanase enzyme prior to stimulation by sFA (mimicking overexpression of the enzyme in obesity-associated breast tumors (29) resulted in 4-fold increased ability of sFA-stimulated macrophages to upregulate ERα expression in E0771 and MCF7 cells (t test p=0.02).”

TLR4 is also expressed on breast cancer cells (PMID: 22042369). Does the enhanced heparinase cleaved HS also have macrophage independent direct effects on the ER expression in the breast cancer cells? It will be useful to show in vitro effects of HS treatment on ER expression in breast tumor cell lines to conclude macrophage specific function.

In Figure 7, the authors should also demonstrate at least one representative IHC image for different grades of ER staining along with heparinase staining.

The authors used the term “normal weight women” in their manuscript. It will be ideal to describe the BMI criteria for this group.

In Fig.3A, the authors did not mention the statistical significance of the densitometric plots. 

The authors have convincingly shown the enhanced tumor growth (PMID: 31481501) and ER expression in the pan hpse-/- mouse. The reviewer would like to know authors perspective on the major source of enhanced heparinase during obesity. 

Inconsistent use of Figure, Fig and figure

Macrophage and 

TAK-242 and TAK242

Author Response

  1. The writing style is not clear. Poorly written, Lengthy sentences and too wordy.

RESPONSE: We made all the possible efforts to respond to the Reviewer's criticisms and improve wording as well as shorten lengthy sentences (e.g., lines 72-74, lines 97-98, etc.)

  1. Do the authors notice any significant difference in tumor growth and ER expression between lean WT and lean hpse-/-mice?

RESPONSE: In the revised manuscript we clearly state that no statistically significant difference was detected between volume of tumors derived from wt lean vs. Hpse-KO lean mice (lines 391-393). Similarly, no significant difference in ER expression levels was noticed between lean WT and lean hpse-/- mice (Fig. 5 of the revised manuscript).

  1. What figure or supplementary data does the authors referring to in following section?  “On the other hand, in vitro treatment with recombinant heparanase enzyme prior to stimulation by sFA (mimicking overexpression of the enzyme in obesity-associated breast tumors (29) resulted in 4-fold increased ability of sFA-stimulated macrophages to upregulate ERα expression in E0771 and MCF7 cells (t test p=0.02).”

RESPONSE: To diminish the repetitive use of bar graphs and WB images throughout the manuscript, in this particular case we preferred to provide all the information obtained in the above-mentioned experiment (including fold increase and p value) in a written form rather than in graphic form.

  1. TLR4 is also expressed on breast cancer cells (PMID: 22042369). Does the enhanced heparinase cleaved HS also have macrophage independent direct effects on the ER expression in the breast cancer cells? It will be useful to show in vitro effects of HS treatment on ER expression in breast tumor cell lines to conclude macrophage specific function.

RESPONSE: As pointed by the reviewer, TLR4 could also be expressed by BC cells (Liao et al., Breast Cancer Res Treat, 2012, 133:853-63). Thus, to conclude macrophage specific function and to rule out possibility that the increase in ERα expression is due to the direct effect of TLR4 ligand (i.e., carry-over via the conditioned medium), we compared ERα protein levels in BC cells which were either untreated, or incubated directly with the canonic TLR4 ligand LPS, or with the medium conditioned by macrophages stimulated by LPS (0.1 ng/ml). Unlike medium conditioned by the LPS-stimulated macrophages, addition of LPS directly to the BC cells did not result in significant increase of ERα levels, confirming macrophage specific function and ruling out the direct effect of BC-expressed TLR4 on ER expression.

This is now clearly stated in the text (lines 311-319 of the revised manuscript) and depicted in the new Suppl. Figure 3.

  1. In Figure 7, the authors should also demonstrate at least one representative IHC image for different grades of ER staining along with heparinase staining.

RESPONSE: as requested, the revised Figure 7 now demonstrates representative IHC image for heparanase staining. Regarding different grades of ER staining: the intensity of staining in all tumor samples was recorded and reported by expert pathologist as weak/moderate/ strong as a part of highly standardized pathological evaluation procedure based on the Guidelines of American Society of Clinical Oncology/College of American Pathologists for Immunohistochemical Testing of Estrogen and Progesterone Receptors in Breast Cancer (as now detailed on lines 122-124 of the revised manuscript). We therefore assumed that it would be more accurate and concise to provide citations for these guidelines and already published representative images of ER staining grades (Refs. 54 and 55 in the revised manuscript), rather than including the actual images in the Figure 7.  

  1. The authors used the term “normal weight women” in their manuscript. It will be ideal to describe the BMI criteria for this group.

RESPONSE: in the revised manuscript we clearly indicate that normal weight women is defined as BMI<25 (line 66)

  1. In Fig.3A, the authors did not mention the statistical significance of the densitometric plots.

RESPONSE: we apologize for not providing this important information in the original version of the manuscript. In the revised version of Fig. 3A we provide densitometric plot based on analysis of 3 independent experiments, and clearly indicate statistically significant increase in ERa expression (p<0.007).

  1. The authors have convincingly shown the enhanced tumor growth (PMID: 31481501) and ER expression in the pan hpse-/-mouse. The reviewer would like to know authors perspective on the major source of enhanced heparinase during obesity. 

RESPONSE: Heparanase expression has been reported in some types of leukocytes, including macrophages (reviewed in Ref. 46 of the manuscript); moreover, several studies utilizing heparanase-deficient macrophages (i.e., Refs. 47-49)  demonstrated that the enzyme expression by macrophages is critically required for their activation and function. Yet, in the setting of breast tumors, carcinoma cells per se appear to represent major cellular source of the enzyme, as repeatedly indicated by numerous observations (reviewed in Ref. 52). Given the above notion and the secreted nature of the enzyme, it is plausible that in the BC under obese conditions the excessive enzyme overexpressed by tumor cells is a main contributor to adverse activation of macrophages. We thank the Reviewer for raising this important point, which is now discussed in lines 406-412 of the revised manuscript.

  1. Inconsistent use of Figure, Fig and figure

Macrophage and 

TAK-242 and TAK242- correct in the paper and figures

RESPONSE: We apologize for these oversights, in the revised version of the manuscript we consistently use terms "Fig.", "TAK-242" and "MÏ•"

Reviewer 3 Report

The paper is of high quality: clear logic, strong molecular evidence, and convincing storytelling. 

There is one major comment: since it is a pathway involving multiple cell lines, it would help that the authors come up with a summarizing pathway map to explain the factors in macrophages and the ones in breast cancer. 

Detailed minor comments: 

[Line 317-319] In both LPS stimulation and TAK242 treatment, there should be another method (e.g. ICC or flow) to quantify cell viability. GapDH bend seems drastically different in figure 3. An (SFA-Mo+TAK), 3. C (LPS-M0) and 3D (Cont). 

Author Response

There is one major comment: since it is a pathway involving multiple cell lines, it would help that the authors come up with a summarizing pathway map to explain the factors in macrophages and the ones in breast cancer. 

RESPONSE: as suggested, we re-written the last paragraph of the Results section (lines 409-417) in an attempt to better summarize specific involvement of the multiple factors (originating in macrophages and breast cancer cells), as well as additional components of obesity-associated BC microenvironment.

Detailed minor comments: 

Line 317-319] In both LPS stimulation and TAK242 treatment, there should be another method (e.g. ICC or flow) to quantify cell viability. GapDH bend seems drastically different in figure 3. An (SFA-Mo+TAK), 3. C (LPS-M0) and 3D (Cont). 

RESPONSE: since macrophages (but not the BC cells per se) were treated with LPS/TAK, we assumed that quantitation of BC cell viability is not obligatory in this experimental setting. In addition, all the data regarding ERa expression had been normalized to GAPDH levels in three independent experiments. We hope the Reviewer will accept this reasoning.

Round 2

Reviewer 2 Report

The Authors have addressed all of my concerns with the original manuscript.

Author Response

The requested data are included in the revised manuscript as new Suppl. Fig. 4